# The Occurrence of the Sensory Processing Disorder in Children Depending on the Type and Time of Delivery: A Pilot Study

**DOI:** 10.3390/ijerph19116893

**Published:** 2022-06-04

**Authors:** Agnieszka Ptak, Diana Miękczyńska, Agnieszka Dębiec-Bąk, Małgorzata Stefańska

**Affiliations:** Physiotherapy Department, University of Health and Sport Sciences in Wrocław, 51-612 Wrocław, Poland; dmiekczynska@gmail.com (D.M.); agnieszka.debiec-bak@awf.wroc.pl (A.D.-B.); malgorzata.stefanska@awf.wroc.pl (M.S.)

**Keywords:** behaviour problems, learning problems, preterm-born, C-section, sensory processing disorder

## Abstract

Over recent years, the concept of Sensory Integration has become more popular. Knowledge about Sensory Processing Disorder (SPD) also has grown, and it is often discussed in scientific research. Sensory disturbances can cause problems in learning and behaviour of children in whom no medical diagnosis has been made. These are healthy children regarding the environment, but their behaviour is often described as strange in the meaning not appropriate/not adequate to the situation. The aim of the study was to analyse if there is a correlation between occurrence of SPD and the time or the way of delivery. Participants were 75 children, ages 5–9 years old. Children born prematurely (*n* = 25), and children delivered by caesarean section (C-section) (*n* = 25) were compared to the ones born on time by natural means (*n* = 25). Research was based on a questionnaire filled by children’s parents. Descriptive results and percentage calculations were compared. SPD were detected among 84% of pre-borns and among 80% of children delivered by C-section and it is statistically significant. Both groups are at higher risk of Sensory Processing Disorder than those delivered on time by vaginal birth. Due to the results, the time and the way of the delivery are the factors that affect Sensory Processing Disorder.

## 1. Developmental Consequences of an Unusual Beginning of Life

The senses provide information about the physiology of the human body and its environment. The development of sensory integration begins in the prenatal period, when the foetus feels the movement of the mother’s body for the first time. The developmental peak occurs between the 3rd and the 6th or 7th year of life [1,2]. The Sensory Processing Disorder (SPD) is a condition characterised by a lack of skills which allow the person to organise the information received by the senses and use it in everyday life [3,4].

The sensory integration dysfunctions can be classified into three categories: the Sensory Modulation Disorder (SMD), the Sensory Discrimination Disorder (SDD) and the Sensory-Based Motor Disorder (SBMD). Those categories can be further divided into the following subtypes: SMD–Sensory Over-Responsivity (SOR), Sensory Under-Responsivity (SUR) and Sensory Craving (SC); SBMD–postural disorder and dyspraxia. The most prevalent category is the SMD [3,4]. Sensory modulation is a basic mechanism of the nervous system that controls the level of stimulation of the brain, as well as human behaviour and emotions. Impaired modulation manifests itself in abnormal reactions to sensory stimuli in the form of attention deficit, excessive mobility, impulsiveness, emotional volatility and inadequate behaviours [5,6]. Such abnormalities may result in difficulties in learning, physical activity, and everyday activities. The social aspect cannot be overlooked. Inadequate sensory responses to peers may not be acceptable to them. Consequently, leading to exclusion from the peer group [7,8].

The causes of the Sensory Processing Disorder include genetic factors, prenatal factors, such as medications, stimulants, complicated pregnancy, multiple pregnancy, preterm delivery, low birth weight, C-section [9], as well as postnatal factors, such as perinatal shock arising from emergency birth or surgery carried out immediately following the birth. Through the senses, the central nervous system learns to respond to stimuli, processing and integrating them. The level of maturity of the nervous system depends on the maturity of the newborn and perinatal events. Despite the excellent neonatal care, a number of sensory anomalies are still observed among babies born prematurely or by C-sectioning. The postnatal factors and circumstances related to the environment in which the baby lives, as well as the manner in which it is cared for and raised, are also of importance. Research shows that prolonged stress resulting from the complications during pregnancy or fear of the C-section may impair the development of the sensory representation of anatomical regions in the sensorimotor cortex, the awareness of one’s own body, cause neurobehavioural disorders, and movement disorders known as dyspraxia, as well as impaired development of visual pathways [10,11,12]. Studies carried out by Bhutta et al. [13] indicate that there are correlations between premature birth, lower intelligence and behavioural issues in school-aged preterm children.

The causes of sensory integration disorders and their impact on the impaired integration of stimuli in children are a subject that is frequently tackled by researchers [14,15]. Under physiologic conditions, the foetal development of a baby lasts from 38 to 42 weeks. Preterm birth interferes with the healthy development of a newborn. Children with more pronounced degrees of preterm birth (gestational age (GA) < 37 weeks) or low birth weight (<2500 g) are at risk for reduced brain volumes in the structures associated with EF, including cerebral white matter, frontal, parietal, and temporal cortices, and the basal ganglia and cerebellum. Not surprisingly, these children have a wide range of deficits in EF compared to term-born normal birth weight (NBW) controls, with deficits proportional to the degree of prematurity [16]. Preterm birth is a factor which adversely influences the functioning of the foetus outside the mother’s body [17,18]. 

The purpose of this article is to assess the type and frequency of the Sensory Processing Disorder occurring in children born prematurely or by C-section. It analyses the relationships between the type of the Sensory Processing Disorder and the frequency thereof, and the type and time of delivery.

## 2. Method

### 2.1. Participants

The study was carried out on three 25-member groups composed of healthy primary school students aged 6–9 (Table 1): Group I—children born spontaneously before the end of the 37th week of gestation; Group II—children born by C-section after the 38th and before the end of the 40th week of gestation; Group III—full-term children born spontaneously (control group). None of the children had a certificate of motor or intellectual disability. None of the children received psychiatric treatment.

Out of 75 subjects, only 12 children (16%) had ever participated in sensory integration activities. The therapy was attended by 6 persons from among the prematurely born children (Group I), 5 persons from among the children born by C-section (Group II), and only 1 person from the control group (Group III).

### 2.2. Research Methods

The study was carried out in January and February 2018 using the Sensorimotor History Questionnaire in the form of a survey for parents of 1st- through 3rd-grade primary school students in the West Pomeranian Voivodeship in Poland.

The Sensorimotor History Questionnaire consists of sections which are used to evaluate the following areas: touch, movement and balance, coordination, muscle tone, hearing, vision, smell and behaviour. All characteristics are assessed with points. Stronger sensory maladjustment is characterised by a higher score and vice versa. The questionnaire is used to assess sensory maladjustment based on the caregiver’s backward observations [19,20]. Based on the literature, the obtained responses were analysed in detail and classified into specific categories of disorders. The presence of sensory modulation disorders is determined by abnormalities in at least three out of five groups of senses (touch, deep feeling, sight, hearing, and taste and smell). Then, within each sense, there was an additional division into three types: hypersensitivity, hypersensitivity or sensory seeking. On the other hand, sensory movement disorders were not subject to an additional classification, and they were identified on the basis of problems with at least three out of five sensory skills (level of activity, feeding, organisation, sleeping, socio-emotional skills) [3,4,8,21]. 

In the first part of the questionnaire, the respondents provided basic information, such as age, gestational age at birth and type of delivery or the child’s participation in the sensory integration therapy. Based on this data, the children were classified into appropriate groups (I, II or III). The second part of the questionnaire concerned the child’s behaviour. The analysis covered the senses and sensorimotor skills in the area of the Sensory Modulation Disorder and the Sensory-Based Motor Disorder. The disarrays were classified based on the analysis of the symptoms occurring in individual subtypes of the Sensory Processing Disorder, according to the classification by Kranowitz [4]. The children were assessed with the Sensory Modulation Disorder based on the occurrence of positive symptoms in at least three out of five groups of senses (touch, movement, vision, hearing, taste and smell).

The statistical analysis was carried out using the Statistica 13.3 programme (Tulsa, OK, USA). The Pearson’s chi-squared test was applied in order to evaluate the statistical significance of the analysed traits.

## 3. Results

As a category of the sensory integration anomalies, the Sensory Modulation Disorder (SMD) is the easiest to diagnose. During the research, in the case of pre-term children, this type of disarray was found in 84% of the subjects from Group I, as compared to the control group where it was assessed only in 12% of the children (Table 2).

The evaluation of the Sensory Modulation Disorder covers the following criteria: touch, movement, vision, hearing, taste and smell. The analysis shows that the maladjustment in all of the areas except for vision are more prevalent in pre-term children than in the control group. Those differences are statistically significant. The most common disarrays were observed in the area of touch and movement and affected 23 and 21 subjects, respectively. The abnormalities related to hearing were observed in 19 subjects, to vision in 14 subjects and to taste and smell in 12 subjects.

The analysis of own research results demonstrates that the Sensory Modulation Disorder (SMD) occurred in 20 out of 25 subjects, which constitutes 80% of the group of children born by C-section. As regards the control group, the abnormalities were found much less frequently and in the present study they affected only three children, i.e., 12% of the group. This difference is statistically significant.

The issues with each of the senses occur statistically significantly more frequently in children born by C-section than in children after vaginal birth. According to the research results, the area affected the most in children born by C-section is hearing, where the disarrays were found in as many as 21 out of 25 subjects. The maladjustation in the areas of movement and touch occurred with comparable frequency, in 19 and 18 subjects, respectively. The maladjustation related to taste and smell was found in 16 persons and the vision-related issues affected the smallest group of children, i.e., 14 subjects.

The results of the statistical analysis show a statistically significant prevalence of disarrays in all evaluated parameters in the group of pre-term children and children born by C-section as compared to the control group, with the exception of vision.

The comparison of both experimental groups during the research did not show a statistically significant difference between the frequency of the Sensory Modulation Disorder in pre-term children and children born by C-section. In both groups, the issues occurred with similar frequency—in 84% of the pre-term children and in 80% of the children born by C-section.

## 4. Discussion

The sensory integration disarrays and the search for their causes are an important subject, frequently investigated by researchers [22]. The analysis of the scientific sources indicates that the gestational age at birth and the birth weight influence the development of the brain [23,24]. As a result, pre-term school-aged children are observed to have lower scores in IQ than full-term school-aged children. The authors have also noticed abnormalities in the area of behaviour in pre-term children as compared to full-term children [16].

Bröring and collaborators prepared a review of the bibliography prepared by Bröring and al., suggesting that so far there has not been enough research that would allow the determination of all possible mechanisms responsible for sensory processing issues and their role in the development of the symptoms. On the other hand, Bröring and co-authors suggest that it can be said with certainty that the Sensory Processing Disorder is found in pre-term children more frequently than in full-term children. This thesis is also corroborated by the results of our research [25].

The analysis of the research results suggests that pre-term children might be considered at risk of sensory integration dysfunctions since such issues were assessed in 84% of the study subjects. In their research, Kranowitz and Miller arrived at the same conclusions [4,11]. Furthermore, the studies carried out by Buffone [26] confirm that the Sensory Processing Disorder occurs more frequently in pre-term children.

This research demonstrates that the Sensory Modulation Disorder in the areas of touch, movement, hearing, taste and smell is statistically significantly more frequent in the pre-term children covered by the evaluation than in the members of the control group. The majority of disarrays affect touch and movement, associated with the vestibular system and proprioception. They constitute the primary sensory systems, on which the processes of integration of sensory stimuli are based, including those related to the three remaining senses [1]. The normal functioning of those systems and the interactions between them are crucial for the child’s behaviour, which—as a socially useful element—is particularly important for the child’s functioning in society. Disorders related to this process are observed both in pre-term children and children with low birth weight [13]. In their work, the researchers frequently describe cases of damage to the central nervous system occurring in pre-term children as those in whom various disorders, such as cerebral palsy or other strongly manifesting conditions, are diagnosed [25]. Finally, there are children who are considered a part of the risk group but have not been diagnosed with any condition; however, their behaviour is described as “strange” (e.g., not possible to sit in one place, need of strong stimuli can be painful) or different from the behaviour of their peers. The researchers should look for the causes of such behaviours.

In our research on the analysed group of children, the Sensory Modulation Disorder manifests itself primarily in the area of touch. This sensory system is the first to develop during foetal life and its functioning is crucial for the growth of the nervous system and its healthy organisation. In the case of pre-term children, the stimulation of the sense of touch is of importance [22]. The results of studies carried out by Als and collaborators indicate that the child’s early experiences affect the functional features and the structure of the brain. If the stimulation of the body by touch is insufficient, the nervous system is not balanced; as a result, the processes of integration of stimuli within the other sensory systems cannot occur properly [22].

Researchers consider birth by C-section a risk factor which predisposes the child to sensory integration disarrays and a factor which does not activate gradual adaptation to the extrauterine conditions or the somatosensory stimulation the child requires [4,11,27]. This research indicates that the Sensory Processing Disorder is found in as much as 80% of children born by C-section and 84% of children born prematurely. This suggests that the children in whom the gestational age at birth and the type of delivery deviate from the physiology of this period are predisposed to the Sensory Modulation Disorder.

The analysis of the research results did not demonstrate any pronounced differences between pre-term children and children born by C-section in terms of the frequency of the Sensory Processing Disorder. In order to consider the differences between the results in both experimental groups and draw conclusions from them, the analysis of a much larger group of children would be required, taking into account the detailed diagnosis of the disarray, and the course of the pregnancy, the perinatal period and early childhood.

## 5. Conclusions

Our research shows that the Sensory Processing Disorder is found in pre-term children and children born by C-section with statistically significantly higher frequency than in full-term vaginally born children.

The present study shows that both the type and time of delivery influence the occurrence of the Sensory Processing Disorder. Sensory integration maladjustations are found in pre-term children and children born by C-section with statistically significantly higher frequency than in full-term baginally born children.

This knowledge can allow the parents and therapists to begin the preventive therapy earlier, based on the medical history; thus avoiding serious developmental deficits. However, this subject requires further research, aimed at explaining the mechanisms responsible for the disarrays in question.

## 6. Limitations

The limitations of this study are small number of participants and no previous detailed medical history of the children as the analysis was made on the basis of a one-time retrospective questionnaire filled in by the parents.

## Figures and Tables

**Table 1 ijerph-19-06893-t001:** Statistical data concerning the subject groups.

	Group I *n* = 25		Group II *n* = 25		Group III *n* = 25		ANOVA
	Mean	SD	Mean	SD	Mean	SD	*p* Value
age (years)	7.52	1.12	7.48	1.05	7.68	0.99	0.7777
gestational age at birth (weeks)	33.72	2.52	39.20	0.76	39.04	0.84	<0.0001 *
	girls	boys	girls	boys	girls	boys	Ch2 test *p* value
sex (*n*)	12	13	12	13	13	12	1.000

* *p* < 0.05.

**Table 2 ijerph-19-06893-t002:** Frequency of the Sensory Modulation Disorder in Groups I, II and III.

		Group			Group		
		I	II	III	I vs. III	II vs. III	I vs. II
		%	%	%	Chi-Square Value		
**SMD**	YES	84	80	12	25.96 *	23.27 *	0.14
touch	over-responsivity	76	56	12	29.13 *	15.92 *	3.54
	craver	16	16	4			
	no	8	28	84			
movement	over-responsivity	20	24	0	17.43 *	13.80 *	0.80
	craver	64	52	28			
	no	16	24	72			
vision	under-responsivity	8	4	0	8.72	8.22 *	0.42
	over-responsivity	28	28	12			
	craver	20	24	12			
	no	0	0	16			
hearing	over-responsivity	44	40	4	18.94 *	23.50 *	0.92
	craver	32	44	12			
	no	24	16	84			
taste and smell	over-responsivity	28	0	8	13.98 *	20.82 *	2.47
	craver	20	40	4			
	no	52	36	88			
**SBMD**	YES	84	72	12	26.03 *	18.47 *	1.92
level of:							
activity	yes	88	80	40	12.50 *	8.33 *	0.60
feeding	yes	68	36	4	22.48 *	8.66 *	5.13 *
organisation	yes	72	68	16	16.16 *	14.31 *	1.03
sleeping	yes	60	44	8	15.06 *	8.42 *	1.28
socio-emotional	yes	84	80	20	20.51 *	18.00 *	0.14
SI therapy	YES	24	20	4	4.15 *	3.03	0.12

* *p* < 0.05.

## Data Availability

The data presented in this study are available on request from the corresponding author.

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
