# Peer review of "The Occurrence of the Sensory Processing Disorder in Children Depending on the Type and Time of Delivery: A Pilot Study"

_ijerph, 2022, doi:10.3390/ijerph19116893_

Round 1
Reviewer 1 Report
Dear author,
thank you very much for having invited me to revise your manuscript.
I have found the topic interesting, however I would like to meliorate the quality of your scientific brief report.
Generally, throughout the manuscript I would avoid statements as “disturb”, “dysfunction” as well.
Mainly, you have collected data from a self-report instrument, as a result you can detect hyper or hypo reaction to some stimuli. Likewise, I would describe all data with caution.
Please, revise the paper as follow:
Introduction:
- Please enrich the introduction adding information by discussion (the two sections should be connected but now they are too similar), the discussion needs to revise.
- Consider explaining the following aspects: medical issues, behaviour influence, developmental lines, neuroscientific issues, correlation with cognitive skills as EF (you can add 4-5 references)
- Please, clarify from 30-40, including better description, some examples and clinical impact.
- Check citation 40° row, double space 52°.
Method:
- Description of groups/participants: order, healthy and medical history information, anamnesis of parents, genetic anomalies, other neuropsychiatric disorders, do someone have followed sensory therapy for a previous clinical issue?
- Table 1. Please report statistics regarding age and sex controls, scholar level (you can use t Student)
- Please, describe what include the therapy followed by some children.
- You need describe the questionnaire adopted: items, sections, examples, psychometric data, articles where it has been implemented. Please report the calculation of your percentages and cut-off of the scales.
Results
- (Table 2.) n°10
- Table 2. You need include Chi-square in the table and * for alpha <0,05 or **/*** and so on. The current table should be simpler. In my opinion, you should report the corrected score showing the cut-off correlated. For example, feeding (cut-off >30); 42, 70%, Chi*. All of your data should be replicable.
- The term “the dysfunction in the area”. Throughout the manuscript you can state “the sensory anomalies”, the hyper reaction.
- Row 13, “prevalence of disorders”, not appropriate term, you do not describe a disability
- Please refer to a total score of the scale and subscale of the instrument in all the results.
Discussion
- “development of the brain” please include e citation as well as “less intelligent” perhaps better low scores in IQ
- “the review of the bibliography”, please extend this concept also in the introduction
- “were diagnosed” inappropriated.
- 169 row double space
- 175, please describe “strange” in TDC as term with some behaviours as example. In all the manuscript.
- 184 rows, check the uncorrected phrase
- In the discussion you need meliorate the language since I confound between citations and data from your current study (Own research…)
- 185 for example you refer to one author, however you have previously cited three researches
- Reading the discussion, it seems you replicate a previous study, don’t you?
- You need to include in the discussion the following sections: interpretation of your results, clinical impact, limitations, advantages, future studies, some clinical innovation
Once again, thank you very much
good luck
Reviewer 2 Report
Can authors clarify they are comparing
1)preterm to term
2)low birth weight to normal range
3) C-section to Vaginal birth.
The conclusion states, "both the type and time of delivery influence the occurrence of the Sensory Processing Disorder. Sensory integration dysfunctions are found in pre-term children and children born by caesarean with statistically significantly higher frequency than in full-term vaginally-delivered children. "
However, preterm birth can result in low birth weight and it is not discussed as a variable.
Authors may consider changing "the both the type and time of delivery influence the occur- 201 rence of the Sensory Processing Disorder. Sensory integration dysfunctions are found in 202 pre-term children and children born by caesarean with statistically significantly higher 203 frequency than in full-term vaginally-delivered children." to neutral language, as this is misleading.
Please provide reference or cite this as authr opinion "both the type and time of delivery influence the occur- 201 rence of the Sensory Processing Disorder. Sensory integration dysfunctions are found in 202 pre-term children and children born by caesarean with statistically significantly higher 203 frequency than in full-term vaginally-delivered children."
